# Extracellular Vesicle lncRNAs as Key Biomolecules for Cell-to-Cell Communication and Circulating Cancer Biomarkers

**DOI:** 10.3390/ncrna10060054

**Published:** 2024-11-05

**Authors:** Panagiotis Papoutsoglou, Antonin Morillon

**Affiliations:** ncRNA, Epigenetics and Genome Fluidity, CNRS UMR3244, Sorbonne Université, PSL University, Institut Curie, Centre de Recherche, F-75248 Paris, France; panagiotis.papoutsoglou@curie.fr

**Keywords:** extracellular vesicles, tumorigenesis, non-coding RNA, biomarkers

## Abstract

Extracellular vesicles (EVs) are secreted by almost every cell type and are considered carriers of active biomolecules, such as nucleic acids, proteins, and lipids. Their content can be uptaken and released into the cytoplasm of recipient cells, thereby inducing gene reprogramming and phenotypic changes in the acceptor cells. Whether the effects of EVs on the physiology of recipient cells are mediated by individual biomolecules or the collective outcome of the total transferred EV content is still under debate. The EV RNA content consists of several types of RNA, such as messenger RNA (mRNA), microRNA (miRNA), and long non-coding RNA (lncRNA), the latter defined as transcripts longer than 200 nucleotides that do not code for proteins but have important established biological functions. This review aims to update our insights on the functional roles of EV and their cargo non-coding RNA during cancer progression, to highlight the utility of EV RNA as novel diagnostic or prognostic biomarkers in cancer, and to tackle the technological advances and limitations for EV RNA identification, integrity assessment, and preservation of its functionality.

## 1. Introduction

### 1.1. Basics of Extracellular Vesicles (EVs)

The term “extracellular vesicles” is an umbrella term to include all the particles defined by a lipid bilayer that cannot be self-divided and are secreted by cells into the extracellular space [1]. The secreted EVs contribute to communication between cells, mainly via the transfer of their content from donor (cells that secrete EVs) to recipient (cells that uptake EVs) cells. The EV content consists of biological macromolecules, such as proteins, RNA, DNA, and lipids, and their release to the cytosol of recipient cells can trigger phenotypic alterations in the acceptor cells. It has been suggested that the EV content highly reflects the cellular content from where they originate. Nevertheless, an integrative proteo-transcriptomic study using matched EV and cell extracts from the same donor cell line revealed that EVs are enriched in distinct protein and transcriptomic signatures compared to cellular counterparts [2].

An important feature of EVs is their heterogeneity in terms of biogenesis site, size, composition of intraluminal content, as well as expression of membranous protein markers. A specific cell type produces and secretes EVs of different sizes and different cargo among individual EVs. Regarding their biosynthetic routes, EVs arise either from the budding of the plasma membrane towards the extracellular space and they are called microvesicles or within multivesicular bodies (MVBs) in the endosomal system, the latter designated as exosomes. Microvesicles have a diameter in the range of 50–1000 nm, whereas exosomes are smaller (50–150 nm) [3]. At this point, it should be noted that according to the Minimal Information for Studies of Extracellular Vesicles 2018 (MISEV2018) recommendations, the use of the term “exosome” does not adequately describe EV identity and, therefore, its use is not endorsed anymore [4]. The use of the term “exosome” to describe specific EV subpopulations in some of the following sections is solely due to the nomenclature used in older studies before the MISEV2018 suggestions, and we did not rephrase it for consistency. Moreover, the use of the term “microvesicle” is currently discouraged by the updated MISEV2023 recommendations. Instead the terms “large EVs” or “small EVs” are favored to describe EVs bigger or smaller than 200 nm in diameter, respectively. 

In terms of EV protein marker expression, different EV populations express different protein markers; however, some protein markers are shared among EVs of different origins and sizes. For example, some markers such as flotillin-1, heat-shock 70 kDa proteins (HSP10/HSP72, HSC70/HSP73), and major histocompatibility complex (MHC) class proteins, previously considered to be found in small EVs (sEVs), have been identified in larger EVs too. The large EVs (lEVs) are highly enriched in the endoplasmic reticulum protein GP96, which is not present in medium-sized or sEVs. The proteins actinin-4 and mitofilin mark the lEVs and medium-sized EVs but not sEVs. The sEVs can be subcategorized further based on distinct protein patterns (mainly the presence or not of different combinations of tetraspanins), as demonstrated by Kowal et al. using EVs from human primary monocyte-derived dendritic cells [5]. The subpopulation that is enriched in the simultaneous expression of CD63, CD9, and CD81 transmembrane proteins, as well as endosomal proteins such as syntenin-1 and tumor susceptibility 101 (TSG101), is considered bona fide exosomes. Although well-defined protein markers are now routinely used to characterize heterogeneous EV populations, we still lack specific RNA markers that could potentially be used to classify EV subpopulations.

### 1.2. A Plethora of Diverse RNA Biotypes Can Be Packaged into EVs

Pervasive transcription of the human genome results in the generation of multiple transcripts, the majority of which are non-protein coding. These non-coding RNAs (ncRNAs) are categorized as short (<200 nucleotides) or long (>200 nucleotides) and function as important regulatory molecules that modulate gene expression. Among the short ncRNAs, different types are classified, such as microRNAs (miRNAs), short interfering RNAs (siRNAs), small nuclear RNAs (snRNAs), small nucleolar RNAs (snoRNAs), and PIWI-interacting RNAs (piRNAs). The group of long ncRNAs includes several subtypes of lncRNAs. A simple classification of the different types of lncRNAs is based on their genomic location with respect to nearby protein-coding genes [6]. Thus, lncRNAs may be bidirectional, antisense, intronic, intergenic, or enhancer RNAs (eRNAs) (Figure 1a). The class of lncRNAs structurally resembles mRNAs in terms of transcription by polymerase II, the addition of methylguanosine cap at the 5′ end and poly-A tail at the 3′ end, and splicing. However, several lncRNAs do not follow this rule and are not considered mRNA-like [7]. They function as regulatory molecules, cooperating with proteins, other types of RNA, or chromatin to modulate gene expression at the epigenetic, co-transcriptional, or post-transcriptional levels [8]. Specific molecular roles as decoys, scaffolds, or guides have been assigned to nuclear lncRNAs that assist or prevent transcription factors or chromatin modifiers from targeting regulatory genomic regions. The cytoplasmic lncRNAs exert mostly post-transcriptional regulatory roles by modulating mRNA stability and translation or by acting as competing endogenous RNAs (ceRNAs) for miRNAs, thereby preventing the latter from interacting with target mRNAs [9]. Another type of ncRNA is the circular RNAs (circRNAs), which derive from the backsplicing of precursor messenger RNA (mRNA), without 5′ or 3′ ends (Figure 1b) [10]. This group of ncRNAs exhibits important regulatory functions by competitively binding to miRNAs, by interacting with proteins, or by being translated into peptides under certain conditions, such as upon m^6^A RNA modification. The translation of circRNAs occurs via a non-canonical route due to the presence of internal ribosome entry sites (IRES) elements in their sequences, thereby omitting the need for 5-cap or the action of translation initiation factors [11].

Several of these different RNA biotypes have been identified in EVs. For example, miRNA is probably the best-studied type of RNA within EVs, with multiple miRNAs being detected in EVs. Transfer RNAs (tRNAs) or fragments of tRNAs are also found to be highly abundant in EVs. A few snoRNAs and snRNAs have been detected in EVs, although with low abundance. Moreover, some piRNAs have also been observed in EVs, although with no apparent biological function. Multiple lncRNAs have been identified in EVs, with attributed molecular and physiological roles, especially in cancer-derived EVs, a topic that will be discussed in more detail in the next chapters. Recently, circRNAs have attracted a lot of interest as they are highly stable transcripts, due to their resistance to degradation by RNAses and their presence in EVs derived from cell lines but also biofluids, rendering them excellent circulating RNA biomarkers. Many more RNA biotypes have been observed within EVs, such as Y-RNAs, vault RNAs, fragments of ribosomal RNA (rRNA), and retrotransposons [12]. Last but not least, mRNAs are also detected in EVs, although their intactness is under debate.

Most of the studies focused on EV transcriptome characterization suggest that EV mRNAs (especially those found in sEVs) are shorter or fragmented compared to cellular mRNAs expressed in the EV donor cell line. Interestingly, 3′-untranslated regions (3′-UTRs) of mRNAs have been described to be preferentially augmented in EVs compared to 5′-UTRs or coding sequences (CDS) [13]. On the contrary, the recent development of robust protocols for long-read sequencing has provided useful evidence for the theory that full-length transcripts can be found in EVs. Indeed, long-read nanopore sequencing in EV and cellular poly-adenylated RNA of human chronic myelogenous leukemia (K562) revealed the presence of full-length EV transcripts (almost 11% of the total EV transcripts identified). Transcripts enriched or depleted in EVs were identified, with lncRNA (42.9%), pseudogenes (35.5%), and mRNA (13.7%) to be the predominant RNA biotypes. Nanopore sequencing offers the advantage of identifying the expression of full-length specific isoforms. Notably, the same study revealed that specific isoforms of the same gene were overrepresented in EVs, compared to cells of the same origin, hindering preferential transcript variant loading into EVs [14]. A second recent study utilizing long-read sequencing (PacBio technology) focused on sEVs from postmortem human brains of patients with Alzheimer’s disease or healthy individuals and demonstrated that sEVs are enriched in mRNAs that encode ribosomal proteins and transposable elements such as human-specific long interspersed nuclear element 1 (LINE-1) (L1Hs). The authors also observed that 80% of the identified neural sEV transcripts were full-length, a finding that challenges many previous reports and will revolutionize the field if it is reproduced in different biological contexts in the future [15].

Another report indicates that long and mature mRNA and lncRNA are found in EVs, using an estimation of their coverage, considering the most 5′ and most 3′ mapping reads within the coding regions of mRNAs or full transcripts of lncRNAs, but not by performing long-read sequencing. The authors observed that EV-associated transcripts are characterized by shorter length but higher exon density compared to cellular transcripts, and in the case of mRNAs, lower frequency of AU-rich elements in their 3-UTR, indicating traits of stable transcripts [16]. Actually, a large cohort of urinary EVs, in comparison to prostate cancer cell lines full transcriptome profiling, using short-read sequencing, also reached the same conclusion, as fully mature mRNA and lncRNA transcripts were identified in sEVs, supporting the idea that sEVs contain a large fraction of intact genetic material [17].

### 1.3. Sorting and Loading of RNAs into EVs: Active or Passive Process?

Several theories for the packaging of RNA into EVs have been developed based on the relative enrichment of a given transcript in EV compared to cellular levels but also on the physicochemical and structural properties of RNA molecules. Evidence for both passive and active RNA loading into EVs exists. Passive loading may take place when RNA is abundant at the sites of EV biogenesis and thus can be spontaneously internalized into EVs. RNA stability is an important factor for passive loading, as more stable cytoplasmic transcripts have more chances to be found in the EV lumen. Active loading includes specific molecular mechanisms for RNA incorporation into EVs. The main determinants for efficient RNA loading are specific RNA motifs and secondary structures, RNA binding proteins (RBPs), modifications in RNA or post-translational modifications of proteins involved in EV RNA packaging, and coupling of EV RNA loading to transcription or translation [18].

A 25-nt RNA sequence, which is called a zipcode motif, has been described in the highly enriched long transcripts found in EVs of brain and skin cancer cells, although this seems to be tissue-specific. In addition, shorter motifs consisting of 4 to 5 nucleotides have been identified in EV miRNAs, such as the motif UGGA, which was found in miRNAs enriched in EVs from MDA-MB-231 breast cancer cells. In a recent study, it was suggested that a part of the 3′-UTR of mRNA RAB13 functions as a signal for the loading of this mRNA into small EVs [19]. A possible mechanism of action for RNA motifs is to establish interactions with lipids within MVBs or plasma membranes at the sites of EV biogenesis, thereby facilitating the loading of the transcripts into newly produced EVs.

The involvement of RBPs in RNA loading into EVs has been better characterized in the case of miRNAs. Several RBPs, such as hnRNPA2B1, hnRNPA1, and hnRNPC, have been shown to interact with specific EV-enriched miRNAs, and this interaction may affect miRNA sorting into EVs [20]. Another RBP, YBX1-forming cytosolic condensates, mediate the sorting of miR-223 into exosomes [21]. Moreover, the RBP FMR1 is implicated in EV miRNA sorting under inflammatory conditions [22]. Interestingly, some non-RBP proteins may also facilitate miRNA sorting into EVs. For instance, connexin 43 (Cx43) may interact with a subset of miRNAs with stable secondary structure elements, i.e., double-stranded hairpin loops, such as miR-133b, and mediate their loading into HEK293 EVs [23]. The mechanisms of long RNA loading into EVs are not yet extensively studied, as in the case of miRNA; nevertheless, a study suggested that hnRNPA2B1 binds to specific motifs in long RNAs (both mRNA and lncRNA) and mediates their sorting into EVs from endothelial cells (HUVECs) [16]. The identified sequences are enriched in GGAG-, UAG-, or GC-contained motifs, which are recognized by classical RBPs, such as RBM5, LIN28A, RBM28, and hnRNPA2B1. Although hnRNPA2B1 has been found in EVs secreted from different cell lines [24], other studies do not confirm this finding. This discrepancy can be explained due to the different EV isolation techniques used in different studies. Unpublished data from our team also supports the absence of hnRNPA2B1 and hnRNPC from MDA-MB-231 secreted sEVs, using size exclusion chromatography (SEC) for EV isolation.

Regarding the precise mechanism by which an RBP facilitates RNA loading into EVs, several hypotheses have been suggested. For example, an RBP may locate transcripts at the sites of EV biogenesis and therefore indirectly affect their incorporation into EVs. Alternatively, RBPs may bind both target RNAs and proteins at the EV membrane, e.g., tetraspanins, and directly mediate EV RNA loading. In the case of miR-223, which was described above, the miRNA is retained within mitochondria due to its interaction with the mitochondrial protein YBAP1. Upon binding to YBX1, it is translocated in cytosolic condensates that ultimately sort it into endosome-derived exosomes [25].

### 1.4. Functionality and Fate of Transferred EV RNA in Recipient Cells

The presence of RNA in EVs is undoubted, as it has been confirmed by several studies; however, the potential biological role of the EV RNA on recipient cells has been the center of debate for decades. The skepticism towards the functionality of EV RNA stems from two main arguments. First, it stems from the idea that EVs contain content (not only RNA but also proteins) that cells deposit for extracellular disposal. Second, the identification of mainly short RNAs or fragments of long RNAs in EVs, using the current RNA extraction protocols and short-read sequencing technologies, led to the notion that full-length mRNAs or lncRNAs are not dominating in EVs and therefore the functionality of the RNA fragments is questionable. Even in the case of miRNAs, the best-described class of ncRNA found in EVs, the identification of high levels of EV-miRNA does not necessarily mean that adequate copies of a specific miRNA are delivered to recipient cells to exert its function towards mRNA degradation or inhibition of mRNA translation.

Another caveat on the capability of transferred EV RNA to exert physiological effects on recipient cells stems from the fact that the delivered EV RNA should escape the degradation pathways in the endocytic compartments of acceptor cells [26]. Since EV uptake by recipient cells is mediated by diverse pathways, such as endocytosis (clathrin or caveolin-dependent), micropinocytosis, phagocytosis, and lipid raft-mediated internalization [27], a large part of delivered EV transcripts need to exit from the endocytic routes that lead to lysosomes and travel towards the cytosol, where they can be translated into proteins or directly function as ncRNAs. Different mechanisms for endosomal escape have been suggested. First, the direct fusion of EVs with the plasma membrane of recipient cells, thereby avoiding encountering endosomes. For those EVs that enter into recipient cells through endocytic routes, it has been proposed that a portion of them escape endosome entrapment via back-fusion [28] or pH buffering [29] and ultimately release their content in the cytosol. Indeed, an elegant study, utilizing luciferase- or fluorescent-protein tagged cytosolic EV cargoes, demonstrated that only approximately 30% of the uptaken EVs are able to release their content to the cytosol, and endosomal acidification plays a crucial role in efficient cytosolic release [30]. Nevertheless, a part of delivered EV RNA will survive decay and eventually will be released to the cytosol of recipient cells. It remains a question of whether the released transcripts have adequate time to exert their functions and shape the responses of recipient cells. A few studies convincingly provide evidence for the functional role of EV-transferred mRNAs, which can be translated into recipient cells. Engineered cells ectopically expressing a bioluminescent reporter encoding the Gaussia luciferase B (GlucB) gene produced EVs enriched in GlucB mRNA. Purified GlucB EVs were uptaken by recipient cells, and nascent mRNA translation was observed in recipient cells by luciferase assays and verified by inhibiting protein synthesis using cycloheximide [31]. Moreover, functional EV mRNA delivery was shown between mouse brain cells in vivo, using brain-derived EVs enriched in Cre mRNA and monitoring the functional release of Cre mRNA with a fluorescent reporter gene expressing tdTomato only upon translation of transferred Cre mRNA in recipient cells [32].

Although the above reports prove efficient delivery of functional mRNA via EVs, they are based on overexpression systems, whereby a gene is constantly expressed in donor cells, and therefore the transcribed mRNA is passively loaded into EVs. However, caution should be shown using these approaches, as an overexpressed protein may also be secreted in the cell-conditioned medium when found in high copies, with the possibility to also be packaged in EVs together with its mRNA. This may lead to false positive results, as shown in a study [33]. To overcome this obstacle, the authors developed an assay whereby they specifically block the production of nanoluciferase protein in donor cells, leaving intact the produced nanoluciferase mRNA, which could be enriched in EVs. By using guide RNAs and RNA editing tools, they subsequently correct the RNA sequence in the recipient cells so that the delivered nanoluciferase mRNA can give rise to nascent nanoluciferase protein using the translation machinery of recipient cells [33].

### 1.5. EVs Possess Multiple Functional Roles During Cancer Progression

The roles of EVs in cancer progression have been extensively studied and reviewed over the years. For efficient growth, tumors develop a complex microenvironment in their vicinity that fuels them with sufficient nutrients and growth factors but also enables them to escape immune surveillance. The tumor microenvironment (TME) consists of non-cancerous cells of different origins, which constantly exchange information with cancer cells through signaling networks. To this point, EVs are considered one of the most important carriers of messages between the different cell types within the TME [34]. Cancer-secreted EVs can be uptaken by immune cells, such as monocytes or dendritic cells, or by fibroblasts and modulate their physiological responses to favor tumor progression. For example, in monocytes, EVs from cancer cells can trigger differentiation towards different populations of macrophages, such as M1 or M2 macrophages. Cancer EVs can also elicit immune suppression via repressing dendritic cells, which are important for antigen presentation and activation of immune responses against cancer antigens. Cancer EVs can also promote the activation of quiescent fibroblasts to cancer-associated fibroblasts (CAFs) via the transfer of growth factors, such as the transforming growth factor β (TGFβ), miRNAs, or mutant p53. CAFs support tumor growth by secreting components of extracellular matrix (ECM) and cytokines (Figure 2). The intercellular communication between cancer and TME cells is not unilateral. Secreted EVs from TME cells can also affect the physiology of cancer cells to further sustain tumor progression. Cancer EVs may play important roles also in the process of metastasis. In locally invading tumors, EVs are secreted at high rates at invadopodia, protrusions of the plasma membrane at the migratory front that allow cancer cells to degrade ECM and therefore promote invasiveness. In addition, at the metastatic sites, EVs can promote metastatic colonization [35].

## 2. EV-ncRNAs with Functional Relevance in Cancer

### 2.1. EV-ncRNAs in Breast Cancer

Several examples of lncRNAs transferred through EVs and with functional consequences in breast cancer progression have been described. For instance, in human epidermal growth factor receptor 2 (HER2) positive breast cancers, the Linc00969 lncRNA has been found to promote resistance to trastuzumab, a monoclonal antibody that targets HER2 expressed in cancer cells. Exosomes isolated from the plasma of patients that developed resistance to trastuzumab were found to have elevated Linc00969 levels, compared to sensitive patients. In addition, in vitro studies confirmed that Linc00969 enriched in exosomes can induce trastuzumab resistance by stabilizing HER2 mRNA levels via binding to the RNA-binding protein HuR and by enhancing autophagy [36]. In another study, the small nucleolar RNA host gene 12 (SNHG12) was identified to be highly expressed in triple-negative breast cancer cells (TNBCs) and also in their secreted exosomes. Exosomes isolated from the MDA-MB-231 TNBC cell line boosted the proliferation, migration, and tube formation capacity of human umbilical vein endothelial cells (HUVECs). Using gain and loss of function experiments, the authors demonstrated that SNHG12 lncRNA packaged in exosomes can be transferred to HUVECs and induce their angiogenesis. To exert its functions in recipient cells, SNHG12 forms ribonucleoprotein complexes with PBRM1 protein to induce MMP10 up-regulation, thereby eliciting angiogenesis [37].

Breast cancer progression is often shaped by the crosstalk between cancer cells and non-cancerous cells of the TME, such as macrophages and fibroblasts. An extensive signaling network between cells of the TME and cancer cells is established to facilitate a favorable environment for efficient tumor growth. Signals originate from fibroblasts or immune cells and are transferred to cancer cells (and vice versa) partially via EVs but also through other pathways. Indeed, tumor-associated macrophages (TAMs) secrete EVs-containing lncRNA HIF-1α-stabilizing long non-coding RNA (HISLA) and potentiate aerobic glycolysis and resistance to apoptosis in recipient breast cancer cells by blocking the interaction of PHD2 and HIF-1α, thereby stabilizing HIF-1α [38]. In another study, EVs secreted by hypoxia-induced tumor-associated fibroblasts contain lncRNA H19, which can be delivered to recipient breast cancer cells and cause a reduction of miR-497 levels in a DNMT1-dependent manner. This molecular function of EV-contained H19 has also physiological consequences for the recipient breast cancer cells as it promotes their growth and resistance to paclitaxel [39].

### 2.2. EV-ncRNAs in Prostate Cancer

The role of EV-lncRNAs in prostate cancer is also well documented. In a study in which a transcriptomic analysis was performed using EV RNA and matched cellular RNA from four prostate cancer cell lines (VCaP, LNCaP, DU145, and PC3), lncRNAs with specific miRNA seed regions in their sequences were enriched in exosomes. Interestingly, the corresponding miRNAs that match with the seed regions of the exosomal lncRNAs were also found highly abundant in the same exosomes. The top identified EV miRNAs belonged to the let-7 and miR-17 families [40]. In a different study, the expression of lncRNA AY927529 is elevated in prostate cancer cell lines and their produced exosomes, compared to human benign prostatic hyperplasia cells [41]. Moreover, the lncRNA NORAD is highly expressed in prostate cancer cells and tissues and potentiates the secretion and uptake of prostate cancer EVs in recipient cells via regulating the miR-541-3p-pyruvate kinase M2 (PKM2) axis [42]. RNA sequencing using EVs in serum from castration-resistant patients with prostate cancer before or after acquiring resistance to second-generation androgen receptor (AR) axis-targeted therapy (ARAT) revealed up-regulation of AR signaling-related lncRNAs, such as PCAT1, H19, HOXA-11AS, ZEB1-AS1, ARLNC1, PART1, CTBP1-AS, and PCA3. Among them, H19 expressed in EVs may negatively correlate with AR-signaling and could serve as a marker for the diagnosis of ARAT resistance [43].

EV RNA content has also been utilized for diagnostic purposes in prostate cancer. For instance, the transcriptome of EVs isolated from urine samples from 20 prostate cancer patients was compared to the urinary EV RNA transcriptome from 9 healthy individuals, and it was observed that five miRNAs (miR-196a-5p, miR-34a-5p, miR-143-3p, miR-501-3p, and miR-92a-1-5p) were downregulated in prostate cancer exosomes. Using an independent cohort, the authors further confirmed the decrease of miR-196a-5p and miR-501-3p in prostate cancer exosomes [44]. The first study to compare the transcriptomic profiling of human prostate cancer tissue to the urinary EVs transcriptome from the same patients revealed that urinary EVs are enriched in shorter and intronless cytoplasmic transcripts compared to the tissue of origin. Notably, circRNA and lncRNA dominate the RNA population in these prostate cancer EVs. Validation of the transcriptomic data from patients using in vitro prostate cancer cell lines identified a few circRNAs with biological function in prostate cancer cells and lncRNAs that may encode peptides with potential tumorigenic function [17].

### 2.3. EV-ncRNAs in Liver Cancer

Hepatocellular carcinoma (HCC) is the most common type of primary liver cancer, associated with high mortality due to its asymptomatic nature at the early stages of the disease. The roles of EV lncRNAs in HCC progression, drug resistance, and metastasis have been elucidated over the last few years. Among the numerous lncRNAs described to be enriched in HCC EVs, the antisense RNA of SLC16A1 (SLC16A1-AS1) is transferred via HCC-secreted EVs to macrophages present in the TME, whereby it promotes M2 macrophage polarization. M2 macrophages secreted the cytokine IL6, which, in turn, boosted proliferation, invasion, and glycolysis of HCC cells, suggesting that EV-enriched SLC16A1-AS1 is a functional mediator of the communication between cancer and TME cells [45]. Moreover, the lncRNA focally amplified lncRNA on chromosome 1 (FAL1) is highly enriched in EVs isolated from the serum of patients with HCC, and treatment of macrophages with FAL1-enriched EVs leads to their polarization towards M2 macrophages, with pro-tumorigenic properties. Co-culturing EV-induced M2 macrophages with HepG2 HCC cells boosted proliferation, stem cell properties, and chemoresistance of the latter through activating the Wnt/β-catenin pathway [46]. In addition, the long intergenic non-coding RNA (lincRNA) ROR (linc-ROR) is enriched in HCC EVs in response to stimulation with the cytokine TGFβ. Moreover, treatment with sorafenib, a chemotherapeutic drug commonly used in HCC, also increased the cellular and EV expression of linc-ROR. Uptake of HCC EVs with high linc-ROR expression by recipient cells led to enhanced chemoresistance of HepG2 cells to sorafenib treatment [47].

### 2.4. EV-ncRNAs in Pancreatic Cancer

In pancreatic cancer, an aggressive type of cancer characterized by poor prognosis, EV lncRNAs with major roles in the progression of this disease have also been identified. In the TME of pancreatic ductal adenocarcinoma (PDAC), CAFs secrete EVs enriched in the lncRNA RP11-161H23.5 that contribute to immune evasion by limiting the expression of HLA-A in PDAC cells. Mechanistically, RP11-161H23.5 interacts with the CNOT4 subunit of the CCR4-NOT complex and promotes the destabilization of HLA-A mRNA by deadenylating its poly-A tail [48]. The lncRNA NNT-AS1 is found to be highly expressed in exosomes from CAFs and to be delivered to PDAC cells, where it acts as a ceRNA for mir-889-3p to stabilize the expression of hypoxia-inducible factor-1 (HIF-1), thereby enhancing anaerobic glycolysis and PDAC progression [49]. The lncRNA XIST (lncXIST), which has well-established functions towards X chromosome inactivation, has been identified in EVs from pancreatic cancer cells, and it is delivered in recipient neural cells, thereby facilitating perineural invasion. Concerning its molecular mechanism of action, lncXIST is a ceRNA for miR-211-5p, the latter targeting glial-cell-line-derived neurotrophic factor (GDNF), which positively regulates perineural invasion [50]. Another EV-enriched lncRNA with implications in PDAC progression is LINC01133. LINC01133 was found to be highly expressed in exosomes secreted by PDAC cells and positively correlated with the poor overall survival of patients with PDAC [51].

### 2.5. EV-ncRNAs in Lung Cancer

Lung cancer is grouped into small cell lung cancer (SCLC) and non-small cell lung cancer (NSCLC), with the latter representing the vast majority of lung cancer cases. Several lncRNAs enriched in EVs have been identified to contribute to lung cancer progression [52]. Exosomes secreted by NSCLC cells contain SOX2 overlapping transcript (SOX2-OT) and promote macrophage M2 polarization, mediated by the ceRNA activity of SOX2-OT, which sponges miR-627-3p, thereby enhancing Smad expression [53]. Another study identified the oncogenic lncRNA LINC00482 to be enriched in serum-derived EVs from patients with NSCLC. EV-enriched LINC00482 induced microglial M2 polarization in vitro by interacting with miR-142-3p and up-regulating TGFβ1 levels. Using an in vivo mouse xenograft model, the authors observed that EVs that contain LINC00482 potentiated brain metastasis of NSCLC [54]. Additionally, M2 macrophage-derived EVs contain lncRNA NORAD, which is delivered to acceptor NSCLC cells to boost their proliferation by binding to miR-520g-3p, thereby stabilizing small integral membrane protein 22 (SMIM22) and UDP-galactose-4-epimerase (GALE) [55]. In a similar way to LINC00482 functions, LINC00313, a lncRNA with pro-tumorigenic functions in several cancer types [56], was identified as overrepresented in NSCLC exosomes, and its exosome-mediated delivery to recipient macrophages promoted their differentiation towards the M2 subtype [57].

### 2.6. EV-ncRNA in Brain Cancer

Brain tumors can be developed in distinct areas of the brain and it is a highly heterogeneous disease with over a hundred different subtypes, depending on the cell type where it arises. In gliomas, glioma stem cells (GSCs) under hypoxia secrete EVs highly enriched in Linc01060, which is then delivered to glioma cells. In glioma cells, Linc01060 promotes myeloid zinc finger 1 (MZF1) stabilization, enhanced c-Myc activity, and increased HIF1α levels, which in turn activate Linc01060 transcriptionally in a positive feedback loop [58]. LncRNA HOTAIR was identified in EVs isolated from serum samples of patients with glioblastoma multiforme (GBM). When serum EVs were added to GBM cells, they increased GBM cell proliferation, invasion, and resistance to temozolomide. At the molecular level, EV-delivered HOTAIR acted as ceRNA for miR-526b-3p, preventing it from binding its target epithelial V-like antigen 1 (EVA1) mRNA [59]. Notably, small peptides translated from small open reading frames (smORFs) within lncRNA sequences have also been identified in glioma cell-derived EVs. These findings were further strengthened by the fact that numerous microproteins were found in plasma-derived EVs from patients with glioma, paving the way to consider circulating lncRNA-encoded small peptides within EVs as disease biomarkers [60].

### 2.7. EV-ncRNA in Colorectal Cancer

Colorectal cancer was the third most frequently diagnosed cancer and ranked second in cancer-associated deaths worldwide in 2022 [61]. In colorectal cancer (CRC), lncRNA WEE2-AS1, enriched in sEVs secreted by CAFs, boosts CRC cell proliferation in vitro. Regarding its molecular mode of action, WEE2-AS1 acts as a scaffold for MOB1A and E3 ubiquitin–protein ligase complexes to facilitate MOB1A degradation, thereby inhibiting Hippo signaling. Notably, plasma-derived sEVs from patients with CRC are characterized by higher WEE2-AS1 levels compared to plasma sEVs from healthy individuals [62]. Another example of a CAF EV-associated lncRNA is SNHG3, which can be delivered into CRC cells, promoting HOXC6 in a miR-34b-5p/HuR-dependent pathway. The ceRNA function towards miR-34b-5p elicits CRC cell proliferation [63]. In another report, the circRNA circ-0034880 was enriched in EVs from patients with CRC, isolated from plasma, and contributed to the establishment of a pre-metastatic niche in the liver. In the liver, circ-0034880-EVs are uptaken by pro-tumor macrophages and transform the TME to favor metastasis [64]. A different role for an anti-tumorigenic circRNA enriched in CRC EVs has been attributed. The circRHOBTB3 levels were found diminished in CRC patient tissues but highly expressed in serum-derived EVs. This circRNA possesses tumor suppressor activities in CRC by interfering with metabolic pathways and reactive oxygen species (ROS) production, and it is suggested to be actively secreted outside cancer cells via EVs to maintain their tumorigenic features [65]. However, the fate of secreted EV-circRHOBTB3 is not explored in the study. Furthermore, a study demonstrated differential sorting of long RNAs in EVs secreted by CRC cells, with a preferential presence of antisense transcripts and pseudogenes in EVs compared to cellular compartments. The authors also confirmed the functional delivery of lncRNAs via EVs to recipient cells using a CRISPR/Cas9-based RNA tracking assay [66].

## 3. EV ncRNA in Cancer Diagnosis and Therapeutics

### 3.1. EV ncRNA as Biomarkers in Cancer

An emerging area in the field of cancer diagnosis is the early detection of cancer with non-invasive methods, especially for solid tumors that are more challenging to perform tissue biopsies. The idea of identifying cancer-specific biomarkers in biofluids of patients, such as blood (plasma or serum) and urine, is appealing, as it can provide a rapid assessment of the expression of specific to each type of cancer transcripts or proteins. However, this is not a trivial task, as molecules that may serve as biomarkers for cancer have to be identified and validated, and the most appropriate type of human biofluid for detection should be determined. Speaking of this, circulating EVs secreted from tumor tissues have the potential to serve as excellent carriers of RNA or protein biomarkers, as their cargo is protected from degradation by the EV lipid membrane [67]. Nevertheless, the challenge of identifying circulating EV RNA biomarkers specific to the tumor remains, as blood contains EVs secreted from different tissues of origin, and, thus, a direct comparison of the expression levels of a biomarker to the bulk tumor tissue is mandatory. Therefore, for particular cancer types, such as prostate cancer, urine collection after prostate massage may represent a better source of EVs that bear cancer-specific biomarkers, which are not diluted in the pool of EVs from different origins within blood. In fact, urine contains EVs (uEVs) mainly derived from kidney, urothelium, and prostate tissues, and since its collection can be scaled up easily, and more importantly, is minimally invasive, it represents an attractive source for studying uEV content for biomarker identification [68]. In the case of cancers of the oral cavity and throat, saliva serves as a rich source of EVs bearing potential diagnostic biomarkers [69]. However, to date, there is not an established salivary EV lncRNA biomarker for oral or throat cancer diagnosis [70]. Just a few miRNAs (miR-302b-3p, miR-517b-3p, miR-512-3p, and miR-412-3p) have been detected in salivary EVs from patients with oral squamous cell carcinoma (OSCC) [71], but again, the robustness of ncRNAs as candidate biomarkers should be further confirmed [72].

The use of liquid biopsies for identifying EV RNA biomarkers in breast cancer is still in its infancy. However, in a study that aimed to elucidate potential RNA biomarkers comparing plasma EVs from 32 patients with locally advanced breast cancer when diagnosed or 7 days post-surgery and also with 30 healthy individuals, the authors constructed a signature of eight RNAs (SNORD3H, SNORD1C, SNORA74D, miR-224-5p, piR-32949, lnc-IFT-122-2, lnc-C9orf50-4, and lnc-FAM122C-3) capable of discerning breast cancer from healthy EVs [73]. In a different study, the same group investigated the potential use of circulating EV RNA for predicting the response of patients with breast cancer to neoadjuvant chemotherapy (NAC). RNA sequencing revealed that 6 miRNAs, 4 lncRNAs (lnc-ALX1-2, lnc-KLF17-1, lnc-DPH7-1, and lnc-PARP8-6), and 1 snoRNA were expressed at higher levels in EVs from non-responders than responders at the time of diagnosis and throughout the period of NAC, and significantly lower levels in healthy individuals, thereby representing promising biomarker candidates for the prediction of response to NAC [74].

A comparison of the RNA content of plasma and urinary EVs from 10 patients with prostate cancer before or after prostatectomy led to the identification of novel RNA biomarkers whose expression is elevated in tumor tissues and EVs before operation but also lost in EVs after patients’ surgery. Interestingly, 63 mRNAs, 3 lncRNAs (Linc00662, CHASERR, and lnc-LTBP3-11), 2 miRNAs (miR375-3p and miR92a-1-5p), and 1 piRNA (piR-28004) were found to be highly expressed in prostate cancer tissues and diminished in urinary EVs post-prostatectomy. This study also confirmed that the use of urinary EVs is superior to the use of plasma EVs for the identification of prostate cancer RNA biomarkers, as plasma contains a mixture of EV populations secreted by different tissues, while urine is enriched in prostate cancer-derived EVs [75].

In bladder cancer, a few lncRNAs have been identified as enriched in patients’ urinary EVs. The lncRNA telomerase RNA component (TERC) was enriched in urinary EVs isolated from patients with bladder urothelial carcinoma (BLCA), representing a potential diagnostic and prognostic biomarker in BLCA [76]. Moreover, lncRNA SNHG16 levels were significantly higher in EVs collected from the urines of 42 patients with bladder cancer compared to 42 healthy individuals [77]. Screening for novel urinary EV lncRNA candidates for bladder cancer diagnosis identified MALAT1, PCAT-1, and SPRY4-IT1 differential expression to be of high diagnostic value in a training set consisting of 208 urine samples, which was subsequently confirmed in a validation set (160 urine samples) [78].

In kidney cancer, a recent study utilized in silico analysis to construct an exosome-related lncRNA score, predicting survival and immunotherapy responses in clear cell renal cell carcinoma (ccRCC). A score consisting of four lncRNAs (EMX2OS, AC026401.3, AC018690.1, and AL161935.1) was determined, and patients with higher scores presented a worse prognosis compared to those with lower scores [79]. However, since the above study is based solely on bioinformatic analysis, the findings should be validated experimentally too. Studies using EVs from biofluids of patients with kidney cancer have focused mainly on identifying EV-associated miRNAs [80]. There is still a lack of studies on the identification of specific EV-lncRNAs with potential use as biomarkers for kidney cancer.

In HCC, circulating exosomes isolated from the serum of 79 patients contained lncRNA-ATB and miRNA-21. High expression of exosomal lncRNA-ATB and miRNA-21 correlated with worse outcomes for these patients, as well as larger tumor size and increased C-reactive protein levels, suggesting that these two ncRNAs have prognostic value in HCC [81]. In another study, lncRNA LINC00853 is overrepresented in sEVs derived from sera of patients with HCC, showing potent discriminatory capacity for early HCC diagnosis with high sensitivity and specificity [82].

In pancreatic cancer, liquid biopsy was performed to isolate EVs from serum samples derived from 20 patients with PDAC, 22 patients with intraductal papillary mucinous neoplasms, and 21 healthy individuals and subjected to lncRNA profiling. A highly expressed EV lncRNA designated as HEVEPA was identified among serums of patients with PDAC but not in the other two control groups [83]. In an independent study, EV-enriched lncRNA HULC was shown to function as a circulating biomarker for PDAC, as it was overexpressed in serum samples from 20 patients with PDAC in comparison to 22 patients with intraductal papillary mucinous neoplasm (IPMN) and 21 healthy individuals [84]. Collectively, the presented examples of functional lncRNAs or circRNAs identified in EVs from cancer cell lines or patient biofluids are grouped in Table 1, according to cancer and sample types, where reported.

Overall, several EV-enriched lncRNAs isolated mainly from patient blood or urine samples have been suggested as candidate circulating biomarkers in certain types of cancer (Figure 3), but it would require further confirmation using a robust large-scale validation cohort of patients from multicenter studies.

### 3.2. EVs as Delivery Tools for Therapeutic Purposes

Drug delivery systems using synthetic nanocarriers are gaining interest in the field of targeted therapy, including cancer. Synthetic lipid nanoparticles can be loaded with small molecules and can be engineered to target specifically cancer cells, thereby minimizing undesirable side effects. Nonetheless, there are still important challenges to face while using synthetic drug delivery systems, such as increased cytotoxicity, reduced biocompatibility, and triggered immunogenicity [85]. The unique properties of EVs related to the protective lipid bilayer that safeguards their cargo, enhanced biocompatibility with minor host immune reactions, and tissue penetrating capacity make them attractive alternative candidates for drug or oligonucleotide delivery-based therapeutic approaches [86]. Indeed, engineered EVs, genetically or chemically modified to express certain biomolecules in their membrane or lumen, have been used in cancer therapy either to specifically target cancer cells or to educate the host immune system against tumor-specific antigens. Synthetic miRNA mimics loaded into EVs are able to prevent migration and self-renewal of glioma cells upon their delivery. Notably, dendritic cell-derived EVs, bearing membrane proteins important for antigen presentation, have been utilized in phase I clinical trials in certain cancers, such as melanoma and colorectal cancer [87].

Apart from cancer, EVs have been used in therapeutics for other diseases. For example, liver fibrosis is characterized by ECM accumulation in the liver tissue in response to chronic damage and can potentially lead to HCC development. To combat the progression of fibrosis, exosomes from mesenchymal stem cells loaded with exogenous siRNA or antisense oligonucleotides (ASOs) against the transcription factor STAT3, a positive regulator of this disease, have been used in a mouse model of liver fibrosis. These engineered EVs efficiently repressed STAT3 and ameliorated liver health [88]. Although the advances in EV therapeutics open up exciting directions for precision medicine in the future, continuous improvement is needed in multiple aspects of this field, mainly in relation to selecting the most appropriate EV donors, enhancing the efficiency of exogenous cargo loading, as well as improving the specificity of target destinations.

## 4. Future Perspectives

The field of EV transcriptomics has been boosted over the last years, mainly due to the improvements in old and development of new EV isolation protocols and also in technological advances in RNA sequencing methods. Nevertheless, a lot of open questions exist and the future research directions in the field include the areas of decoding EV heterogeneity by developing single EV RNA sequencing [89], investigating EV RNA integrity using long-read sequencing, and unraveling the roles of EVs in TME by establishing cancer cell 3D cultures and organoids to better recapitulate the cancer–TME crosstalk. Most of the in vitro studies on cancer cell EVs make use of 2D cultures, a model that does not mimic the structural complexity of tumors in vivo. On the contrary, the development of 3D cultures or spheroids derived from cancer cell lines shortens the distance between conventional cancer cell culture conditions and cancer tissue. Recent reports suggest that the cargo composition of EVs derived from 3D cultures is different from the 2D culture of the same cell line. In addition, 3D cultures seem to enhance EV secretion compared to 2D cultures [90].

Another important task in the field is to better understand whether the EV-mediated effects on recipient cells are the collective outcome of the full cargo transfer or attributed to a few individual biomolecules that are biologically functional. In order to elucidate this possibility, a deconvolution of EV cargo biomolecules is necessary, with the development of EV RNA tracing approaches to monitor the routes and fate of EV single RNA molecules from EV biogenesis in donor cells till their delivery into the cytoplasm of recipient cells. From the cancer biomarker perspective, blood, serum, and urine biofluids are gold-standard sources of EV RNA biomarker identification; however, EVs are secreted in 20 different human biofluids, a finding that opens up possibilities for unraveling novel biomarkers for different disorders or elucidating the functions of EVs in various human physiological processes in the future [91].

## 5. Conclusions

Although both lncRNA and EVs were considered “junk genomic regions” and “carriers of unneeded cellular components destined for disposal,” respectively, when they were first discovered, the scientific community now appreciates their unique biological functions towards cancer progression and their utility in cancer diagnosis as biomarkers and in therapeutics as natural nanocarriers. This fact highlights the need for a meticulous delineation of the functions of newly identified molecules or biological entities before undermining their biological significance, as they may hold promising potential for biological systems.

## Figures and Tables

**Figure 1 ncrna-10-00054-f001:**
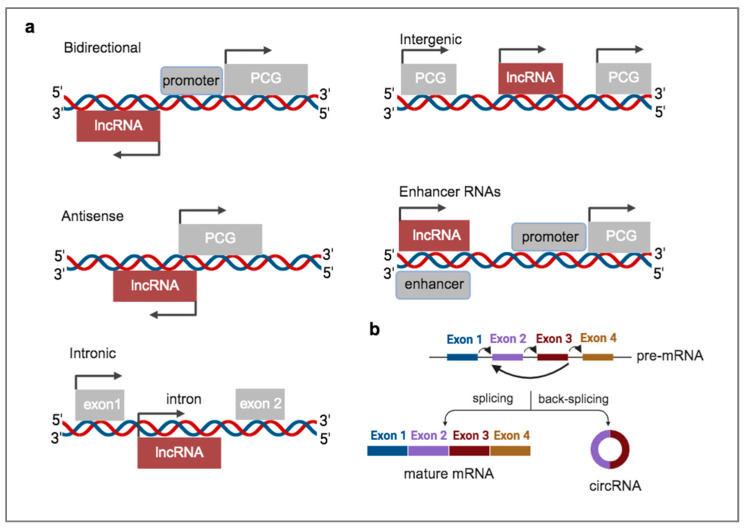
Different subtypes of lncRNAs and circRNA biogenesis. (**a**) Classification of lncRNAs based on their genomic location relative to protein-coding genes (PCG); (**b**) generation of circRNA from back-splicing of a precursor mRNA (pre-mRNA). Arrows in panel (**a**) show the direction of transcription and in panel (**b**) splicing events.

**Figure 2 ncrna-10-00054-f002:**
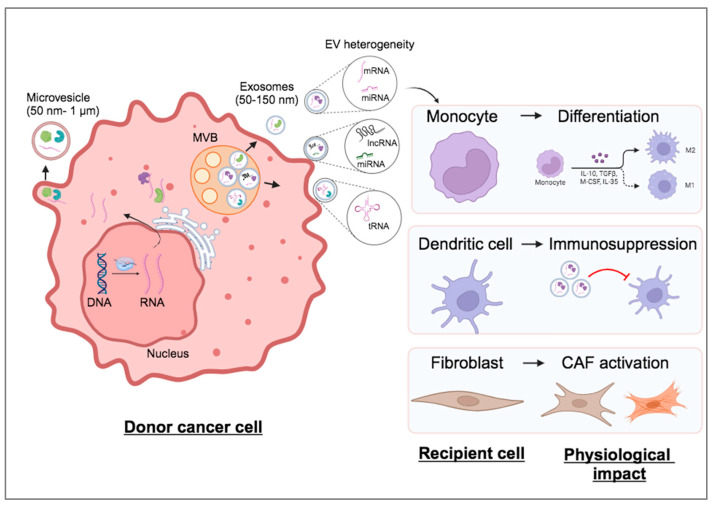
Biological effects of cancer-derived EVs on diverse cell types within TME. Cancer cells produce heterogenous EV populations consisting of different RNA cargo compositions and sizes, which are secreted in the extracellular space and shape the physiological responses of recipient non-tumorigenic cells of the TME, such as monocytes, dendritic cells, and fibroblasts. MVB: multivesicular body.

**Figure 3 ncrna-10-00054-f003:**
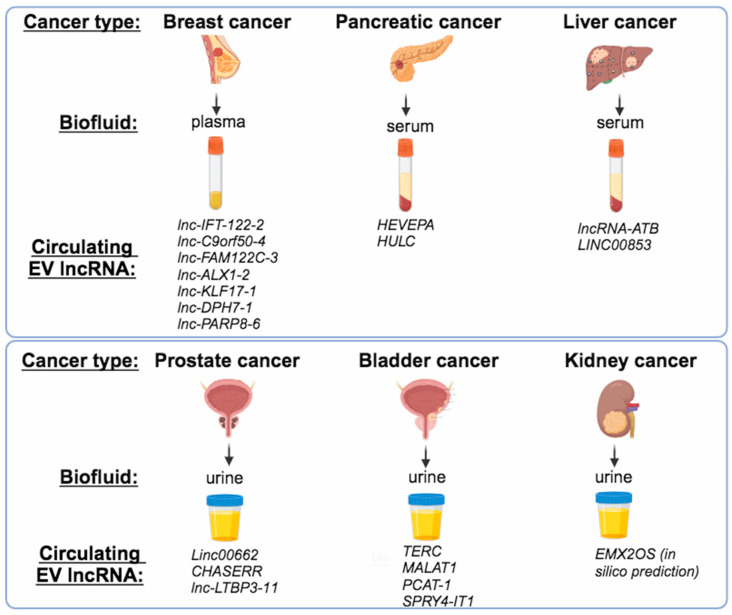
Circulating lncRNAs within EVs as cancer biomarkers. Examples of lncRNAs demonstrated to be highly enriched in EVs from human biofluids derived from patients with breast, prostate, pancreatic, liver, bladder, and kidney cancer.

**Table 1 ncrna-10-00054-t001:** EV-lncRNAs or circRNAs identified in cancer cell lines or patient cohorts.

Cancer Type	lncRNA	Sample Type	Reference
Breast	*Linc00969*	BT474-TR, SKBR-3-TR, and HER2+ serum of patients with breast cancer	[36]
	*SNHG12*	MDA-MB-231 TNBC	[37]
	*HISLA*	tumor-associated macrophages (TAMs)	[38]
	*H19*	tumor-associated fibroblasts	[39]
	*lnc-IFT-122-2*, *lnc-C9orf50-4*, *lnc-FAM122C-3*	Plasma from patients with locally advanced breast cancer or 7 days post-surgery vs. healthy	[73]
	*lnc-ALX1-2*, *lnc-KLF17-1*, *lnc-DPH7-1*, *lnc-PARP8-6*	Plasma from patients with breast cancer with neoadjuvant chemotherapy (responders vs. non-responders)	[74]
Prostate	*AY927529*	VCaP, LNCaP, DU145, PC3	[41]
	*NORAD*	22Rv1, DU145, PC-3	[42]
	*H19*	Serum from patients with CRPC before or after resistance to ARAT	[43]
	*Linc00662*, *CHASERR*, *lnc-LTBP3-11*	Urine from patients with prostate cancer before or after prostatectomy	[75]
Liver	*SLC16A1-AS1*	HepG2, MHCC97H	[45]
	*FAL1*	Serum from patients with HCC	[46]
	*linc-ROR*	HepG2 stimulated with TGFβ	[47]
	*lncRNA-ATB*	Serum from patients with HCC	[81]
	*LINC00853*	Serum from patients with HCC	[82]
Pancreatic	*RP11-161H23.5*	Primary CAFs from patients with PDAC	[48]
	*NNT-AS1*	CAFs from patients with PDAC	[49]
	*lncXIST*	PANC-1, ASPC-1	[50]
	*LINC01133*	SW1990, CFPAC-1, AsPC-1, Panc-1	[51]
	*HEVEPA*	Serum from patients with PDAC vs. patients with IPMN vs. healthy	[83]
	*HULC*	Serum from patients with PDAC vs. patients with IPMN vs. 21 healthy	[84]
Lung	*SOX2-OT*	NSCLC cell line H1975	[53]
	*LINC00482*	NSCLC patient serum or A549 (EV donor), HMC3 (recipient cells)	[54]
	*NORAD*	THP-1 treated with PMA (EV donor cells) and A549 (EV recipient cells)	[55]
	*LINC00313*	NCI-H1299	[57]
Brain	*Linc01060*	Primary tumor cells and GSC from human GBM	[58]
	*HOTAIR*	Serum from patients with GBM	[59]
Colon	*WEE2-AS1*	CAFs	[62]
	*SNHG3*	Primary CAFs	[63]
	*circ-0034880*	Plasma from patients with CRC	[64]
	*circRHOBTB3*	Serum from patients with CRC	[65]
Bladder	*TERC*	Urine (patient vs. healthy)	[76]
	*MALAT1*, *PCAT-1*, *SPRY4-IT1*	Urine (patient vs. healthy)	[78]

## Data Availability

No new data were created or analyzed in this study. Data sharing is not applicable to this article.

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
