# Peer review of "Extracellular Vesicle lncRNAs as Key Biomolecules for Cell-to-Cell Communication and Circulating Cancer Biomarkers"

_ncrna, 2024, doi:10.3390/ncrna10060054_

Round 1

Reviewer 1 Report

Comments and Suggestions for Authors

Dear authors,

The manuscript is well-organized and engagingly written, making it a pleasure to read. To further enhance the manuscript, I suggest 2 minor amendments

(1) Incorporate the discussion on salivary EVs (and their lncRNAs) as a potential diagnostic marker in oral and throat cancers. Sections on urine EVs to be expanded to include bladder and kidney cancers. 

(2) incorporate a table (in lieu of Figure 3) that systematically summarizes the lncRNAs discussed in various cancers and patient cohorts/cell lines. The table should include the cancer type, lncRNA names, sample types, and their relevant references. These additions would provide a clearer overview of the diagnostic (and perhaps prognostic) potential of lncRNAs in cancer, facilitating easier navigation and comprehension of the reviewed literature.

Author Response

Dear authors,

The manuscript is well-organized and engagingly written, making it a pleasure to read. To further enhance the manuscript, I suggest 2 minor amendments

(1) Incorporate the discussion on salivary EVs (and their lncRNAs) as a potential diagnostic marker in oral and throat cancers. Sections on urine EVs to be expanded to include bladder and kidney cancers. 

Authors response

We thank the reviewer for finding our article well-organized and easy to read. We have now included in the relevant section the roles of salivary EVs and their ncRNA cargo as potential diagnostic markers in the cancers of oral cavity. Curating the current literature, we did not find relevant examples of salivary EV lncRNA that could be considered as oral cancer biomarkers. However, there are a few examples of identified salivary EV miRNAs derived from samples of oral cancer patients. We have also added paragraphs focused on urinary EVs for bladder and kidney cancer biomarker detection (subsection 3.1). Although we were able to find several documented urinary EV-lncRNAs for bladder cancer, we did not observe any specific urinary EV-lncRNAs from kidney cancer patients with potential biomarker utility. On the contrary, miRNAs enriched in urinary EVs from renal carcinoma patients have been reported.

(2) incorporate a table (in lieu of Figure 3) that systematically summarizes the lncRNAs discussed in various cancers and patient cohorts/cell lines. The table should include the cancer type, lncRNA names, sample types, and their relevant references. These additions would provide a clearer overview of the diagnostic (and perhaps prognostic) potential of lncRNAs in cancer, facilitating easier navigation and comprehension of the reviewed literature.

Authors response

As suggested by the reviewer, we provide a table (Table 1) that summarizes the identified EV lncRNAs in the different cancer types both for in vitro studies and for those identified using patient cohorts. This indeed facilitates the readers to go through the different presented examples in an easier way. Furthermore, we have also modified the figure 3, in order to include urinary EV-lncRNAs identified in bladder and kidney cancer patients.

Reviewer 2 Report

Comments and Suggestions for Authors

The review manuscript by Papoutsoglou and Morillon nicely described the current state-of-the-art of the role of extracellular vesicle lncRNAs in cancer. The authors nicely provide an overview of the role of extracellular vesicles and the non-coding RNAs, centering thereafter on the role of EV carried lncRNAs in distinct oncogenic processes. The review is very well-written and it provide a very nice and comprehensive overview. There are still two minor points to be addressed. One is that the authors considered circRNAs as a type of lncRNA, To my knowledge this is not fully acepted in the literature and thus I would suggest to reconsider such claim. Secondly, the authors nicely illustrate the role of distinct EV-loaded lncRNAs in different cancer types, including breast, prostate, liver and pancreatic cancer. I wonder if there are no information in other cancer types such as glioma or lung cancer or it is just that the authors highlighted only a subset of these oncogenic processes. I would be important to provide a rationale of why only these types are highlighted.

Author Response

The review manuscript by Papoutsoglou and Morillon nicely described the current state-of-the-art of the role of extracellular vesicle lncRNAs in cancer. The authors nicely provide an overview of the role of extracellular vesicles and the non-coding RNAs, centering thereafter on the role of EV carried lncRNAs in distinct oncogenic processes. The review is very well-written and it provide a very nice and comprehensive overview. There are still two minor points to be addressed. One is that the authors considered circRNAs as a type of lncRNA, To my knowledge this is not fully acepted in the literature and thus I would suggest to reconsider such claim.

Authors response

We thank the reviewer for finding our manuscript well-written and comprehensive. We understand the reviewer’s point concerning the inclusion of the class of circRNAs into the general class of lncRNAs. The reality is that several reviews categorize circRNAs as a subtype of lncRNAs (such as Mattick et al., 2023), while others consider them as a distinct class of non-coding RNAs, due to their characteristic features, such as the circularity. Our view is that circRNAs have definitely features of ncRNAs and they are longer than 200 nucleotides, which is the arbitrary cut-off chosen by the scientific community to define lncRNAs. However, we appreciate their distinct features that differentiate them from the rest of lncRNAs. Therefore, we have followed the reviewer’s suggestion and we have removed statements that circRNA is a type of lncRNA.

Secondly, the authors nicely illustrate the role of distinct EV-loaded lncRNAs in different cancer types, including breast, prostate, liver and pancreatic cancer. I wonder if there are no information in other cancer types such as glioma or lung cancer or it is just that the authors highlighted only a subset of these oncogenic processes. I would be important to provide a rationale of why only these types are highlighted.

Authors response

Concerning the second part of the comment, the answer is that definitely there are reported examples of EV-ncRNAs with functional roles in additional cancer types, however we chose to focus in the aforementioned four cancer types as examples of solid cancers, whereby the use of circulating EV-associated lncRNAs from liquid biopsies could be useful as non-invasive diagnostic biomarkers. However, it is true that lung and brain cancer could also be included in this list and we have now provided a few examples of EV-lncRNAs with functional impact in these two cancer types (subsections 2.5 and 2.6) and we have also included colorectal cancer EV-associated lncRNAs (subsection 2.7). However, for the sake of space we did not include an exhaustive list of examples for each one of these cancers.